# The Prevalence of Elder Abuse and its Association with Frailty in Elderly Patients at the Outpatient Department of a Super-Tertiary Care Hospital in Northern Thailand

**DOI:** 10.3390/medicina59091644

**Published:** 2023-09-11

**Authors:** Yanee Choksomngam, Terdsak Petrungjarern, Perapoln Ketkit, Pakpoom Boontak, Ratchanon Panya, Tinakon Wongpakaran, Nahathai Wongpakaran, Peerasak Lerttrakarnnon

**Affiliations:** 1Department of Family Medicine, Faculty of Medicine, Chiang Mai University, Chiang Mai 50200, Thailand; yanee.choksomngam@cmu.ac.th; 2Faculty of Medicine, Chiang Mai University, Chiang Mai 50200, Thailand; terdsak_p@cmu.ac.th (T.P.); perapoln_ketkit@cmu.ac.th (P.K.); ratchanon_panya@cmu.ac.th (R.P.); 3Department of Psychiatry, Faculty of Medicine, Chiang Mai University, Chiang Mai 50200, Thailand; tinakon.w@cmu.ac.th (T.W.); nahathai.wongpakaran@cmu.ac.th (N.W.)

**Keywords:** elder abuse, violence, psychological abuse, frailty, super-tertiary care

## Abstract

*Background and Objectives:* The global population is undergoing rapid aging, resulting in an increase in geriatric syndromes and hidden health issues such as elder abuse. However, the prevalence of elder abuse varies across different settings. The objective of this study is to determine the prevalence and factors associated with elder abuse at the outpatient department of a super-tertiary care hospital in northern Thailand. *Materials and Methods:* This cross-sectional study involved 210 elderly patients who visited Maharaj Nakorn Chiang Mai Hospital between May and August 2022. The participants completed several assessments, including the Mini-Cog, Thai Geriatric Depression Scale-6, Core Symptom Index-15, FRAIL scale, Barthel Activities of Daily Living, Interview Guideline for Screening for Elder Abuse, and Diagnostic Criteria for Elder Abuse. Fisher’s exact test was used to examine the association between the factors and elder abuse. *Results:* The Screening for Elder Abuse yielded noteworthy results, with 15.7% of the elderly patients having experienced psychological abuse. However, only a smaller subset of study participants, comprising five individuals or 2.38%, met the diagnostic criteria for elder abuse. Furthermore, statistical analysis revealed no significant correlation between elder abuse and the other factors examined in the study. *Conclusions:* As a result, it is crucial for hospitals to consider preventive measures and implement routine screening protocols.

## 1. Introduction

The global population is undergoing rapid aging. According to the data from the United Nations Department of Economic and Social Affairs, the global population of individuals aged 65 and older reached 727 million in 2020, and it is projected to exceed 1.5 billion by 2050 [1]. Similarly, in Thailand, there is an increasing proportion of elderly individuals. The “Situation of the Thai Elderly 2020” report from the Foundation of Thai Gerontology Research and Development Institute indicates that in 2010, there were 1.7 million elderly individuals, accounting for 4.9% of the total population. By 2020, the number of elderly individuals had risen to 12 million, representing 18.1% of the total population. Projections suggest that by 2040, there will be 20.5 million elderly individuals, comprising 31.4% of the total population [2]. With the growing number of elderly individuals, the prevalence of elder abuse is also expected to increase [3].

Elder abuse, also referred to as the abuse of older people, encompasses actions or the absence of appropriate actions within relationships where trust is expected. These acts or omissions can lead to harm or distress for the elderly individuals involved. Such violence violates human rights and manifests in various forms, including physical, sexual, psychological, emotional, financial, and material abuse [4]. Additionally, it can also involve abandonment, neglect, and a significant loss of dignity and respect [5]. Abuse of older people can have serious consequences, including premature mortality, physical injuries, depression, cognitive decline, poverty, and placement in long-term care institutions [6,7,8].

A systematic review conducted in 2017, analyzing 52 studies from 28 countries across various regions, provided valuable insights into the prevalence of elder abuse in community settings. The findings revealed that approximately 15.7% of individuals aged 60 years and older had experienced some form of abuse within the past year [9]. Notably, the prevalence of elder abuse in institutional settings was found to be even higher. Reports from staff members indicated an overall estimate of 64.2% who admitted to engaging in elder abuse within the past year. When older residents themselves reported on their experiences, the prevalence rates for abuse subtypes were highest for psychological abuse at 33.4%, followed by physical abuse at 14.1%, financial abuse at 13.8%, neglect at 11.6%, and sexual abuse at 1.9% [10]. A China-based study conducted in 2018 found that older individuals with chronic diseases were vulnerable to neglect in various aspects of their lives, including life neglect, medical neglect, and neglect in their residential environment. The study also revealed that older individuals with better cognitive abilities were less likely to experience neglect, while a decline in their daily activity capacity was associated with a higher risk of elder neglect [11].

Additionally, recent evidence from the United States shows that the prevalence of elder abuse has increased by approximately 83.6% during the COVID-19 pandemic compared to pre-pandemic estimates [12]. Similarly, in China, the prevalence of elder abuse has also risen during the COVID-19 epidemic [13]. However, despite the widespread occurrence and severity of elder abuse, it remains a low priority globally. The data on the prevalence of elder abuse in different countries and settings are still limited [14]. Therefore, the United Nations Decade of Healthy Ageing 2021–2030 provides a unique 10-year opportunity to address the issue of elder abuse in a coordinated and sustained manner [4].

In Thailand, a previous study was conducted among elderly individuals living in metropolitan Bangkok that revealed that 14.6% of the participants had experienced elder abuse [15]. The study identified several factors associated with elder abuse, including being female, experiencing poor family relationships, and having a high level of dependency [15]. Furthermore, several studies have highlighted the association between elder abuse and frailty [16,17,18].

Frailty is a condition characterized by a heightened vulnerability to the inability of the body to maintain homeostasis after a stressful event. It is strongly linked to negative outcomes, including falls, worsening disability, hospitalization, institutionalization, and even death [19]. Previous studies have categorized frailty measurements into three approaches: the physical phenotype model of Fried et al., the deficit accumulation model of Rockwood and Mitnitski, and the mixed physical and psychosocial models, such as the Tilburg Frailty Indicator and the Edmonton Frailty Scale. The use of different assessment tools has resulted in varying prevalence rates across healthcare settings and regions [20]. However, a meta-analysis has revealed that frailty is associated with a 1.8–2.3 fold increased risk of mortality, a 1.6–2.0 fold increased risk of loss of activities of daily living, a 1.2–1.8 fold increased risk of hospitalization, a 1.5–2.6 fold increased risk of physical limitation, and a 1.2–2.8 fold increased risk of falls and fractures [21].

Frailty presents a significant challenge in the context of population aging due to its association with severe consequences. Considering the potential impact of frailty on escalating dependency and support requirements [18,19], the aim of this study is to determine the prevalence of elder abuse and its association with frailty in elderly patients attending the outpatient department of a super-tertiary care hospital in Northern Thailand.

## 2. Materials and Methods

### 2.1. Study Design 

This study is a cross-sectional survey conducted between May and August 2022. Ethics approval was obtained from the Research Ethics Committee, Faculty of Medicine, Chiang Mai University (Approval No. 134/2022).

### 2.2. Participants

This study included 210 participants. The inclusion criteria were Thai national patients aged 60 years and older who visited the outpatient department of Maharaj Nakorn Chiang Mai Hospital. Participants with severe dementia or communication problems were excluded from the study. The study was conducted after written consent was obtained from the participants (Figure 1).

### 2.3. Measuring Instruments

As this research topic is sensitive, the interviews were conducted exclusively with the elderly in private locations, with relatives waiting outside. Data were collected on the following aspects: (1)Demographic characteristics, which include sex, age, education, income, marital status, family members, family relationships, and caregivers.(2)Psychological assessments, which were conducted using the Mini-Cog [22], Thai Geriatric Depression Scale-6 [23], and Core Symptom Index-15 [24]. The Mini-Cog serves as a rapid screening tool for early dementia detection, comprising three steps: three-word registration, clock drawing, and three-word recall. Participants received 1 point for each spontaneously recalled word without cues and 2 points for accurately drawing a normal clock. The maximum score is 5 points, with a cognitive impairment cut-off of less than 3 [22]. The Thai Geriatric Depression Scale-6 is a common screening tool for assessing depression in older adults. It consists of 6 questions, each scored from 0 to 1, with a depression cut-off of 2 or higher [23]. The Core Symptom Index-15 is used to evaluate symptoms of anxiety, depression, and somatization. It comprises 15 questions rated on a 5-point Likert scale: 0 (never), 1 (rarely), 2 (sometimes), 3 (frequently), and 4 (almost always). Scores range from 0 to 60, with a psychopathology cut-off of 29 or higher [24].(3)Functional assessment, which includes the FRAIL scale [25] and Barthel Activities of Daily Living [26]. The FRAIL scale is a short frailty screening tool consisting of 5 questions that assess fatigue, resistance, aerobic capacity, illnesses, and weight loss. Each question is scored from 0 to 1, categorizing patients into three groups: normal (score = 0), prefrail (score = 1–2), and frail (score = 3–5) [25]. The Barthel index is an ordinal scale used to measure functional independence in personal care and mobility domains. It comprises 10 questions, each scored from 0 to 2. This index classifies patients into three categories: total dependence (score = 0–4), partial dependence (score = 5–11), and independence (score ≥ 12) [26].(4)Interview guideline for screening for elder abuse and diagnostic criteria for elder abuse [15]. The screening guideline consists of six questions:
Have you not received care that you should have received from relatives, caregivers, or family members?Have you suffered or had difficulties caused by the actions of family members or others?Have you been sad, sorrowful, or disappointed by the actions or expressions of family members or others?Have you been frightened by the actions or expressions of family members or others?Have you been hurt or become sick because of actions or expressions of family members or others?Have you been taken advantage of or been cheated by a family member or others?

If any of the following statements above are answered with a “yes”, continue with the assessment following the diagnostic criteria:An intentional attempt to harm includes physical force, physical coercion, physical or drug-induced restraints, verbal or non-verbal expressions, non-consensual sexual contact, and illegal or improper exploitation.Refusal or failure to fulfill caregiving obligations, whether intentional or unintentional.Either Statement 1 or Statement 2 leads to danger or suffering for the elderly person, including pain, injury, illness, or psychological distress.Statement 1 or Statement 2 is not behavior intended to benefit or protect the elderly person, including health-related benefits or the safety of the elderly person.Statement 1 or Statement 2 originates from a person who has a relationship with the elderly person and is socially expected to be trustworthy in refraining from violent behavior towards the elderly person.

Diagnostic criteria for elder abuse: must meet the criteria of Statement 1 or Statement 2. Additionally, must fulfill all the other remaining criteria: 1 (+3, +4, +5) or 2 (+3, +4, +5).

In cases where participants had abnormal assessment results, researchers provided appropriate care recommendations and any necessary referrals.

### 2.4. Sample Size Calculation

The sample size calculation was based on the primary objective of this study, which aimed to determine the prevalence of elder abuse. To estimate the finite population proportion, with a population of 37,168 (number of patients who visited the outpatient department of Maharaj Nakorn Chiang Mai Hospital in 2021), a proportion (p) of 0.146 (based on a previous study conducted among elderly living in metropolitan Bangkok [15]), an error (d) of 0.05, and an alpha (α) of 0.05, a minimum sample size of 191 was required. In total, 210 patients were included in the study, providing feasible and complete information.

### 2.5. Statistical Analyses

We used STATA version 16 (StataCorp. Statistical Software: Release 16. College Station, TX, USA: StataCorp LLC.) for data analyses. The percentage was used to represent the prevalence of elder abuse. Descriptive statistics and Fisher’s exact test were used to examine the association between factors and elder abuse.

## 3. Results

The mean age of the patients was 70 years; most of them were female (55.24%) and had below basic education (50%). Sixty-one percent of the patients had a monthly income of less than 10,000 Baht, and the majority considered their income to be sufficient (55.71%). Additionally, 69.51% were married, and the majority lived with their spouse and/or children (85.71%). Most of them perceived their family relationships to be good (98.57%), and 87.43% had informal caregivers (Table 1).

### 3.1. Mental and Functional Status

In Table 2, the assessment of mental and functional status showed that 60% had cognitive impairment, 16.66% had depression, 1.90% had abnormal psychological symptoms based on the assessment of the Core Symptom Index, 52.38% had pre-frailty, 15.24% had frailty, and 98.57% had independent daily living activities.

### 3.2. Interview Guideline for Screening for Elder Abuse and Diagnostic Criteria for Elder Abuse

In Table 3, the screening for elder abuse showed that 15.71% were sad, sorrowful or disappointed by actions or expressions of family members or others, 9.05% were taken advantage of or cheated by a family member or others, and 6.67% suffered or had difficulties caused by the actions of family members or others. However, a smaller subset of participants, comprising five individuals or 2.38%, met the diagnostic criteria for elder abuse.

### 3.3. Characteristics of Elderly Individuals Who Have Been Abused

The study did not identify any statistically significant factors associated with the occurrence of elder abuse. Among the elderly individuals who were abused, their average age is 69.2 years, and the majority of them are female. They tend to have lower levels of education and monthly incomes below 10,000 baht, with many perceiving their income as insufficient. Additionally, they do not have formal caregivers and do not have cognitive impairment or depression. Moreover, there is no evidence of frailty among this group. Interestingly, all of them live with their spouses and/or child and have good family relationships, as indicated in Table 4.

## 4. Discussion

Among the 210 elderly patients, 15.7% reported experiencing psychological abuse. However, only five individuals, who accounts for 2.38% of the total, met the diagnostic criteria for elder abuse. The study did not find any statistically significant factors that were correlated with elder abuse.

The prevalence of elder abuse in this study differs from the previous research conducted among elderly individuals living in metropolitan Bangkok, which reported a prevalence rate of 14.6% [15]. Similarly, a systematic review of 52 studies from 28 countries in 2017 found that around 15.7% of individuals aged 60 years and older in community settings had experienced elder abuse in the past year [9]. Additionally, the research conducted in communities in Mexico revealed a prevalence rate of elder abuse of 35.7% [18]. These differences can be attributed to the fact that the previous studies focused on elderly individuals residing in communities, while our study specifically targeted elderly individuals receiving outpatient care at a super-tertiary care hospital. We hypothesized that the majority of elderly individuals visiting the hospital are capable of self-care or have attentive caregivers, which reduces their risk of experiencing abuse. Our results, which support this hypothesis, reveal that 98.57% of participants had independent daily living activities, and 87.43% had informal caregivers. These findings are consistent with previous studies that identified several factors associated with elder abuse, including experiencing poor family relationships and having a high level of dependency [14,15]. Another setting that may play an important role in detection of elder abuse is the Emergency Department [27,28]. However, elder abuse is infrequently detected. A previous study found that elder abuse was diagnosed in only 0.013% of the United States emergency department visits, which is at least two orders of magnitude lower than the estimated prevalence in the population [29]. Furthermore, there are several other studies that have reported various population demographics and definitions of elder abuse, resulting in variations in the reported prevalence rates among the elderly [5,30,31]. 

The lack of statistically significant correlations in this study may be attributed to the low prevalence of elder abuse. However, previous studies conducted in out-of-hospital and emergency department settings have identified several factors that are more commonly associated with elder abuse. These factors include being female, cognitive impairment, functional disability, frailty, social isolation, lower socioeconomic status, and psychiatric and substance abuse disorders [32]. These findings align with a systematic review published in 2013, which identified multiple risk factors for elder abuse, such as cognitive impairment, behavioral problems, psychological issues, functional dependency, frailty, low income, history of trauma or abuse, ethnicity, caregiver burden, caregiver psychological problems, family disharmony, poor or conflictual relationships, low social support, and living arrangements with others [16]. Similarly, a systematic review published in 2019 highlighted impairment/dementia, poor mental health, low income/socioeconomic status, financial dependence, gender, age, and race/ethnicity as commonly cited risk factors for elder abuse [33].

In this study, no correlation was found between elder abuse and frailty, which differs from previous studies [16,17,18,34]. A systematic review identified frailty as one of the multifactorial risk factors for elder abuse [16]. A study conducted in a northwest city of Turkey demonstrated that frailty, abuse, and depression are significant concerns among older adults living in the community, and there are notable connections among these factors [17]. In Mexico, a study revealed an association between frailty and both overall abuse and conflict-related abuse in community-dwelling older adults [18]. In Brazil, a study conducted with hospitalized older adults highlighted frailty as a risk factor for elder abuse [35]. Furthermore, another study reported that indicators of frailty can be used to predict referrals to social workers for evaluating cases of elder abuse within the Veterans Administration healthcare system [34]. These variations in findings may be attributed to the differences in study settings and the use of various assessment tools [19]. A previous study revealed variations in frailty prevalence rates depending on the assessment tool used. For example, the rates were found to be 8% when using the FRAIL scale, 13% with the Fried and modified versions, 17% using the Clinical Frailty Scale, 17% with PRISMA-7, 18% using the Kihon checklist, 30% with the Edmonton Frailty Scales, 38% using the Tilburg Frailty Indicator, 51% with the Groningen Frailty Indicator, and 55% when employing the Comprehensive Geriatric Assessment [36]. However, the Asia-Pacific Clinical Practice Guidelines for the Management of Frailty emphasize that there is currently no universally accepted reference standard for identifying frailty [20]. Similarly, a systematic review published in 2017 stated that it is unable to recommend a single tool because the selection and implementation must be suitable for the specific context [37]. In our study, we opted for the FRAIL scale to assess frailty because this tool was recommended by the Department of Medical Services, Ministry of Public Health, Thailand, as outlined in the recent Thai Handbook of Health Screening and Assessment for the Elderly [38]. 

While the study did not identify any statistically significant factors correlated with elder abuse, it did yield noteworthy results from the elder abuse screening. Specifically, it revealed that 15.7% of the elderly patients had experienced psychological abuse. These findings are consistent with the previous research conducted in both Thailand and internationally. For instance, a study conducted in the community of Bangkok found the highest prevalence of psychological abuse among the elderly, with a rate of 41.18% [15]. Similarly, the research conducted in Romania found the highest prevalence of psychological abuse among the elderly, reaching 60.2% [39]. Furthermore, a study conducted in 25 primary health care centers in Saudi Arabia revealed that 88.3% of participants reported at least one symptom or suspected symptom of psychological elder abuse [40]. The factors that predicted vulnerability to psychological elder abuse included age, gender, income, living arrangements, functioning, and social networking [40]. Additionally, a study conducted in a tertiary care hospital in Sri Lanka found that those who experienced psychological abuse, in comparison to the non-abused, had three or more children and lived in a house belonging to the elderly person [41]. 

While several studies indeed report a higher prevalence of psychological abuse compared to other subtypes, measuring psychological abuse poses significant challenges. Consequently, the healthcare sector plays a crucial role in preventing, raising awareness, and providing evidence-based guidance to healthcare practitioners for addressing psychological abuse [9]. Implementing preventive measures and routine screening protocols to address elder abuse effectively is imperative. Doctors should inquire about the emotional well-being of elderly patients during each hospital visit, asking whether they are experiencing feelings of sadness, sorrow, or disappointment due to the actions or behavior of family members or others.

Another group involved in reporting abuse cases and subsequently managing victims includes clinical forensic practitioners. Their regular medical inspections and medico-legal examinations play a crucial role in the subsequent legal proceedings. Forensic physicians also serve as vital intermediaries among various professionals engaged in identifying and caring for these specific cases, including psychologists, social workers, medical care providers, lawyers, and investigative personnel [42]. Additionally, there is a need for thorough and detailed documentation, along with the careful collection of medico-legal data [42,43]. Therefore, promoting accurate medical records is vital to preventing medical malpractice liability and ensuring adherence to clinical care standards. Implementing evidence-based guidelines and surveillance for maintaining high-quality care and preventing abuse in long-term care for older adults are key public health goals [43].

A strength of this research is that it provides valuable insights into elder abuse in the outpatient department of a super-tertiary care hospital, where there was previously no existing evidence. However, there are limitations regarding the sample size. The researchers determined the sample size based on previous studies conducted on community-dwelling elderly in Thailand, where elder abuse prevalence is higher. This resulted in an insufficient number of participants to identify statistically significant factors. Therefore, it is recommended to increase the sample size in future studies of the elderly in a hospital setting.

Additionally, we have identified interesting points for future research, including:Gathering data on the care mechanisms of elderly individuals receiving outpatient care at a super-tertiary care hospital compared to those residing in the community will help us provide more specific care.Exploring the reasons behind the predominance of psychological abuse will contribute to preventive measures that are context appropriate.

## 5. Conclusions

The prevalence of elder abuse among patients at the outpatient department of a super-tertiary care hospital is 2.38%. However, no statistically significant correlations were found between elder abuse and other related factors. Despite this, the screening process did reveal a significant occurrence of psychological abuse, indicating the importance of implementing preventive measures and routine screening protocols in hospitals to address this issue effectively.

## Data Availability

The data presented in this study are available on request from the corresponding author. The data are not publicly available due to privacy and ethical restrictions.

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
