# Peer review of "The Prevalence of Elder Abuse and its Association with Frailty in Elderly Patients at the Outpatient Department of a Super-Tertiary Care Hospital in Northern Thailand"

_medicina, 2023, doi:10.3390/medicina59091644_

Round 1

Reviewer 1 Report

Thank you for the opportunity to read this well written, clear and concise study. It was interesting and a pleasure to read.

The introduction is very complete and allows the reader to know the main topic of the research, informs about the purpose and importance of the work in the clinical field, and also answers the question posed in the scientific context. It includes previous works on the subject in question and makes clear the detailed aspects of the review, which constitutes the object of the proposed investigation. It explains the general problem of the research, includes previous work on the topic in question, and specifies the objective of the study. In any case, it is necessary to make the following modifications:

- Lines 39 and 45: the same reference cannot be placed in two consecutive paragraphs. Please add a new reference in one of the lines.

- Line 44: after the word "abuse", it is necessary to include a bibliographical reference.

- Line 91: replace the word "objective" with "aim".

The “methods” section is one of the most fundamental sections of a scientific article. Therefore, it must be ordered and completed:

- It must have the following structure:

         -2.1. Study design: in which the type of study and the approval code of the corresponding Ethics Committee must be specified.

        - 2.2. Participants: the total sample participating in the study and the inclusion and exclusion criteria are specified. Also, a participant flowchart should be included.

         - 23. Measuring instruments: not only must the instruments used be named, their use and procedure must be described.

         - 2.4. Sample size calculation

         - 2.5. Statistical analysis used in which the program used is specified.

The authors of this article have meticulously explained the results by providing relevant tables to what is explained in the text. On the other hand, they have provided a clear and complete discussion comparing their results with previous studies and arguing the existing differences. Furthermore, the limitations of this research are presented very clearly.

Likewise, the references section complies with the journal's standards.

Reviewer 2 Report

The presented study on elder abuse offers an invaluable insight into an essential topic, yet there are several areas that merit scrutiny.

- The authors compare their findings to numerous other studies from a myriad of settings. While it is commendable to situate one's study within the broader literature, these comparisons might be of limited relevance given the distinct nature of the populations being studied. Specifically, comparing outpatient elderly individuals at a Super-Tertiary Care Hospital to community-dwelling elderly or those in different geographic or socio-economic settings might lead to imprecise interpretations. Please refer also to studies and review focused on Hospital or Healthcare facilities such as Rosen et al., Clin Geriatr Med. 2018; Cimino-Fiallos and Rosen, Emerg Med Clin North Am. 2021; Gallione et al., J Clin Nurs. 2017; Franceschetti et al., For Sci International 2022; D’Anna et al., Healthcare (Basel) 2023.

- In addition, the discrepancy between this study's findings on frailty and previous research is noteworthy. The inconsistent findings on frailty's role in elder abuse could be due to various definitions and assessments of frailty across different studies, which the authors do touch upon. Still, it might be worth delving deeper into how frailty was quantified and assessed in this study compared to others.

- Furthermore, the focus on psychological abuse in the findings is significant, but the study might have benefitted from a more in-depth exploration of why this type of abuse was predominant. Are there specific conditions within the outpatient context that make psychological abuse more prevalent?

- The hypothesis that "the majority of elderly individuals visiting the hospital are capable of self-care or have attentive caregivers" is plausible. However, this remains a supposition without clear empirical backing. If this was a central tenet of the research, it would have been pertinent to gather data on the care mechanisms these individuals have in place or their general health and dependency status.

- The article notes that there was a lack of statistically significant correlations with elder abuse. However, the emphasis seems to be more on statistical significance than clinical or practical significance. It would have been useful to understand and better explained if any non-statistically significant trends were observed and their potential implications.

- The manuscript points out that this research is the "first study in Thailand to investigate the prevalence and factors associated with elder abuse in the outpatient department of a super-tertiary care hospital.” While it is crucial to understand the unique contributions of a study, it's equally essential to not overemphasize its novelty if the methodology and sample size have clear limitations.

- The conclusion rightly underscores the importance of routine screening protocols. However, the article might have further benefited from recommendations on how hospitals and other care settings can specifically address the high incidence of psychological abuse.

In conclusion, while the study contributes to the burgeoning literature on elder abuse, its findings should be interpreted with caution, given its methodological limitations and contextual disparities with other referenced studies. Future research in this area would benefit from a more comprehensive exploration of the outpatient context, ensuring a representative sample size, and potentially employing mixed methods to capture a holistic understanding of elder abuse dynamics.

The language used is generally clear, coherent, and academically appropriate. The use of technical and professional vocabulary is consistent with the standards of scientific writing. The structure of sentences and the flow of ideas from one paragraph to the next exhibit a logical and organized progression.

However, there are minor stylistic and grammatical inconsistencies. For instance, the term "su-per-tertiary" is hyphenated in a manner that may not be standard, and it would be preferable to see it as "super-tertiary." Additionally, while the article often successfully integrates data from various studies, the transitions between different pieces of evidence can occasionally feel abrupt. A smoother integration or synthesis of such studies, perhaps using transitional phrases, could enhance the flow and clarity of the article.

In conclusion, the English used in the article is commendably professional and mostly free from glaring errors. However, slight refinements in style and grammar could further elevate the quality of the presentation.

Round 2

Reviewer 2 Report

The manuscript gained in consistency after authors made the requested revisions. The article is quite ready for publication. However, they did not add all the suggested references, without an explanation: when considering older abuse, do the Authors think the forensic and medico-legal aspects should not be considered?
